# Swelling-Resistant, Crosslinked Polyvinyl Alcohol Membranes with High ZIF-8 Nanofiller Loadings as Effective Solid Electrolytes for Alkaline Fuel Cells

**DOI:** 10.3390/nano12050865

**Published:** 2022-03-04

**Authors:** Po-Ya Hsu, Ting-Yu Hu, Selvaraj Rajesh Kumar, Kevin C.-W. Wu, Shingjiang Jessie Lue

**Affiliations:** 1Department of Chemical and Materials Engineering, Chang Gung University, Guishan District, Taoyuan City 333, Taiwan; mini50636@gmail.com (P.-Y.H.); hu70308@gmail.com (T.-Y.H.); rajeshkumarnst@gmail.com (S.R.K.); 2Department of Chemical Engineering, National Taiwan University, Taipei City 106, Taiwan; 3Institute of Biomedical Engineering and Nanomedicine, National Health Research Institutes, Miaoli County 350, Taiwan; 4Department of Safety, Health and Environment Engineering, Ming Chi University of Technology, Taishan, New Taipei City 243, Taiwan; 5Division of Joint Reconstruction, Department of Orthopedics, Chang Gung Memorial Hospital, Linkou, Taoyuan City 333, Taiwan

**Keywords:** polymeric electrolytes, ZIF-8 nanofillers, glutaraldehyde, alkaline stability, direct alkaline methanol fuel cell

## Abstract

The present work investigates the direct mixing of aqueous zeolitic imidazolate framework-8 (ZIF-8) suspension into a polyvinyl alcohol (PVA) and crosslinked with glutaraldehyde (GA) to form swelling-resistant, mechanically robust and conductivity retentive composite membranes. This drying-free nanofiller incorporation method enhances the homogeneous ZIF-8 distributions in the PVA/ZIF-8/GA composites to overcome the nanofiller aggregation problem in the mixed matrix membranes. Various ZIF-8 concentrations (25.4, 40.5 and 45.4 wt.%) are used to study the suitability of the resulting GA-crosslinked composites for direct alkaline methanol fuel cell (DAMFC). Surface morphological analysis confirmed homogeneous ZIF-8 particle distribution in the GA-crosslinked composites with a defect- and crack-free structure. The increased ionic conductivity (21% higher than the ZIF-free base material) and suppressed alcohol permeability (94% lower from the base material) of PVA/40.5%ZIF-8/GA resulted in the highest selectivity among the prepared composites. In addition, the GA-crosslinked composites’ selectivity increased to 1.5–2 times that of those without crosslink. Moreover, the ZIF-8 nanofillers improved the mechanical strength and alkaline stability of the composites. This was due to the negligible volume swelling ratio (<1.4%) of high (>40%) ZIF-8-loaded composites. After 168 h of alkaline treatment, the PVA/40.5%ZIF-8/GA composite had almost negligible ionic conductivity loss (0.19%) compared with the initial material. The maximum power density (P_max_) of PVA/40.5%ZIF-8/GA composite was 190.5 mW cm^−2^ at 60 °C, an increase of 181% from the PVA/GA membrane. Moreover, the P_max_ of PVA/40.5%ZIF-8/GA was 10% higher than that without GA crosslinking. These swelling-resistant and stable solid electrolytes are promising in alkaline fuel cell applications.

## 1. Introduction

Increased environmental awareness of the need for low-polluting gas emissions makes fuel cells a popular alternative energy to fossil oil [1]. Accordingly, various renewable-energy-related systems have been developed, including solar cells, supercapacitors, metal-ion/metal-air batteries, and fuel cells, that have been recognized as potential energy storage devices [2,3]. Among these, the proton-exchange membrane (PEM) and anionic-exchange membrane (AEM) fuel cell technologies are appropriate for portable devices and energy power supply [4,5]. However, PEM fuel cells have limitations in mass production. The high costs of PEM membranes and metal electro-catalysts are major concerns regarding commercialization [4]. Alternatively, the alkaline fuel cells are beneficial for favorable kinetic reactions at both the cathode and the anode [6,7,8,9]. Cost-effective catalysts, such as non-noble metals, metal oxides, and carbonaceous nanocomposites, can be used as effective electro-catalysts [10,11,12]. Moreover, an alkaline solution is advantageous for higher power density (P_max_) due to the higher catalytic efficiency and a higher rate of fuel oxidation process than that of acidic medium [13]. Therefore, alkaline fuel cells are more economically feasible than the proton-exchange counterpart.

Direct methanol fuel cells (DMFC) are studied next to hydrogen fuel cells due to their low cost and high supply quantity. It was reported that the methanol oxidation rate in the alkali medium is faster than in acidic environment [14]. The opposite transport of OH^−^ ions from the cathode region into the anode section can suppress methanol cross-over. A significant improvement in the water management system (as water is produced at anode compartment) alleviates the cathode flooding issue. Moreover, non-platinum catalyst materials are available in the DMFC [15].

Although many AEMs are developed for alkaline fuel cells, there are still challenges to overcome. The compromise between ionic conductivity and fuel cross-over is the main concern. High conductivity and low fuel permeability are unlikely to be achieved at the same time. We reported that mixed matrix membranes (MMMs) with nanofillers in polymers are an easy approach to simultaneously obtain highly conductive composites to retard fuel cross-over [16,17]. Among these polymers, polyvinyl alcohol (PVA) is a nontoxic synthetic polymer, exhibits favorable reactivity for crosslinking, has a more hydrophilic nature, and possesses eco-friendly characteristics. The ease of film formation and the ability to separate water from alcohol [18] also make these materials suitable for use as AEMs [19]. Moreover, the abundant OH groups in the polymer side chain give PVA a high affinity for water. Alkaline-doped PVA exhibits suppressed permeability and increased ionic conductivity due to the increased amorphous region after filler incorporation. This was characterized in suitable free volume, which allows water permeation but inhibits large alcohol molecules [14,16]. Researchers have recently paid more attention to improving the P_max_ for DMFC by optimizing the catalyst loadings, effects of temperature, alcohol concentrations and acidic vs. alkaline doped electrolytes. Some typical literature results are summarized in Appendix A.

The mechanical strength of PVA membranes always becomes poor after long-term operation because of swelling in aqueous solutions [20]. It has been demonstrated that the existence of a crosslinker into the PVA polymeric matrix improves the mechanical strength [21,22]. Furthermore, Diaz et al. [23] reported that after a post-crosslinked polyvinyl alcohol/polybenzimadazole (PVA/PBI) composite was immersed in KOH solution for 7 days, there was no change in the membrane except for the color, while the un-crosslinked PVA/PBI membrane became brittle. Merle et al. [22] proposed that pre-crosslinked PVA maintained its integrity and did not experience a color change when doped in 1 M potassium hydroxide (KOH) at ambient temperature for one month, demonstrating the high tolerance of this material to an alkaline environment.

However, this crosslinked polymer becomes mechanically stable at the expense of its ionic conductivity. PVA reacts with glutaraldehyde (GA) crosslinker, and the hydroxyl groups in PVA are consumed [24]. In addition, the PVA structure becomes constrained, restricting molecules from diffusing and reducing fuel crossover through the membrane. That also implies less water and KOH adsorption in the polymer [23]. Rudra [25] et al. prepared pre-crosslinked PVA with GA, and the results found that the water uptake and degree of membrane swelling reduced as compared with un-crosslinked samples. In our previous research, the quaternized PVA (QPVA)/GO-Fe_3_O_4_ composites were placed in the GA crosslinking solution and the post-crosslinked QPVA membrane exhibited a lower solubility in hot water at 60 °C than the pre-crosslinked membrane (i.e., GA in PVA solution before cast and drying). The suppression of swelling in this post-crosslinked QPVA confirmed the membrane stability in water. The ionic conductivity of the GA-crosslinked QPVA/GO-Fe_3_O_4_ composite was 0.0076–0.0108 S cm^−1^ [26], which was drastically lower than those of the un-crosslinked QPVA/GO-Fe_3_O_4_ composite membranes (0.0468–0.0548 S cm^−1^) [27]. Decreased direct alkaline methanol fuel cells (DAMFC) performance was observed using the GA-crosslinked QPVA/GO-Fe_3_O_4_ membrane [26]. We suspect that nanofiller aggregation in the QPVA/GO-Fe_3_O_4_ composite membrane induced crack formation between the polymer and fillers, which caused voids and defects, leading to undesirable properties [18,28,29]. 

Homogeneous mixtures are required to prepare MMMs with good compatibility between the fillers and polymers. Deng et al. [30] developed a water-based technique to synthesize ZIF-8 particles and mix ZIF-8 aqueous slurry with PVA solution to promote good nanofillers dispersion in PVA matrices without particle aggregates. Using that protocol, we report on the presence of porous ZIF-8 nanofillers in a polymeric matrix that improves the OH^−^ conductivity and suppresses alcohol permeability. However, the composite membrane stability in the KOH solution is not sufficient over long-term operations. The ionic conductivity decreased by ~32% after the composite was placed in a 6 M KOH solution for 168 h [31].

We attempted to crosslink the PVA/ZIF-8 composite with homogeneous ZIF-8 nanofiller distribution. This is performed by mixing ZIF-8 suspension and an aqueous PVA solution and crosslinked with GA for alkaline stability. The membrane characteristics of the pre-crosslinked PVA/ZIF-8/GA composites with various ZIF-8 loads are studied. The selectivities of GA-crosslinked composites were greater than those of the un-crosslinked samples. The stability in alkaline, volume swelling ratio, ionic conductivity, and mechanical strength of these composites are improved in crosslinked composites. The DAMFC performance using the proposed composites with various ZIF-8 loads is evaluated, and the effects of filler load and GA crosslinking are discussed. 

## 2. Materials and Methods

### 2.1. Materials

Zinc nitrate hexahydrate (Zn(NO_3_)_2_ 6H_2_O) was obtained from Acros Organics (Morris Plains, NJ, USA). Polyvinyl alcohol (PVA, 99% hydrolyzed, 146–186 k Da), 2-methylimidazole, potassium hydroxide (KOH), hydrochloric acid (HCl, 37%) glutaraldehyde (GA), Nafion, and methanol were acquired from Sigma-Aldrich (St. Louis, MO, USA). 

### 2.2. Preparation of GA-Crosslinked PVA/ZIF-8 Composites 

2-methylimidazole (18 g) and Zn(NO_3_)_2_ 6H_2_O (1 g) were mixed in 60 mL of deionized (DI) water to form ZIF-8 suspension through the water-based coprecipitation method. The aqueous PVA solution was added with 25.4% of ZIF-8 particle suspensions to form PVA/ZIF-8 mixed solutions [30]. A 0.25 g crosslinker solution (1.5 wt.% GA, 1 wt.% sulfuric acid, and 97.5 wt.% water) was mixed with the above solution under continuous stirring for 30 min. Then, the solution was cast on the surface of the glass substrate using an application knife (clearance of 600 μm). The PVA/ZIF-8/GA composite was placed in a vacuum oven and dried at 80 °C. Different concentrations (40.5 and 45.4 wt.%) of ZIF-8 nanofillers were prepared in the PVA polymeric matrix using a similar procedure. The thickness of the dried sample was measured to be 40–45 µm.

### 2.3. Physical-Chemical Properties

The morphological properties of PVA/GA and PVA/ZIF-8/GA composites were studied using field emission scanning electron microscopy (FESEM, model JSM-7500F, Hitachi High-Technologies Corp., Tokyo, Japan) and transmission electron microscope (TEM, model TECNAI-20, New York, NY, USA). The elemental composition was determined using energy dispersive X-ray spectroscopy (EDS) attached with FESEM and X-ray photoelectron spectroscopy (Thermo Scientific™ K-Alpha™ XPS, Thermo Fisher Scientific, Waltham, MA, USA). The specific surface area and pore size distribution of the ZIF-8 nanoparticle were examined using nitrogen adsorption-desorption isotherm measurements (Micomeritics ASAP2020, Norcross, GA, USA). The phase purity of the PVA/GA and PVA/ZIF-8/GA composites was determined using X-ray diffraction (XRD, model D5005D, Siemens AG, Munich, Germany). The chemical structure of the samples was determined using Fourier transform infrared spectroscopy (FTIR, Model Spectrum 100, Perkin- Elmer Inc., Shelton, CT, USA). The degree of crystallinities was estimated by means of differential scanning calorimetry (DSC, Perkin- Elmer Inc., Shelton, CT, USA) under a nitrogen atmosphere. The polymeric crystallinity was calculated using the below equation

$$\chi_{C}=\frac{\Delta H}{\Delta H_{C}\left(1-\Phi \right)}$$

where 
ΔH
 is the melting enthalpy data of the polymeric composite, 
$\Delta H_{c}$
 represents the melting enthalpy, and 
Φ
 is the loading amount (in wt.%) of ZIF-8 nanofillers in the composite membrane [32]. The mechanical behaviors of the composites were analyzed using a tensile-strength tester (model AI-3000, Gotech Testing Machines Inc., Taichung, Taiwan) at ambient temperature [33]. Elongation at break; tensile strength; and Young’s modulus were reported. The alkaline uptakes, swelling ratios, ionic conductivities, and methanol permeabilities were measured according to previous procedures [14,16,29,31,34] after the composites were doped with 6 M KOH solution to form hydroxide-conductive electrolytes.

### 2.4. Single-Cell Measurement

The experimental setup for single-cell measurement was detailed in our previous work [14,16,29,31,34]. To evaluate the cell performance, the PVA/GA/ZIF-8 composites were dried and placed in a 6 M alkali solution for at least 12 h. The 1.5 × 1.5 cm^2^ PVA/GA/ZIF-8 composites were placed between two gas diffusion electrodes (GDEs) to set up a membrane electrode assembly (MEA) with a 1 cm^2^ active area. The GDEs consisted of carbon cloth sprayed with Pt-Ru/C (2 mg cm^2^) for the anode and Pt/C (1 mg cm^2^) for the GDE cathode, with the catalyst layers facing the electrolyte membrane. To prevent liquid fuel leakage, Teflon gaskets were fixed between the carved flow field plates and surrounded the MEA. Conductive end plates (thickness of 10 mm) were placed next to the flow field plates [14]. The 2 M methanol/6 M KOH solution was used as the anode feed (at a flow rate of 5 mL min^−1^). Humidified oxygen gas was fed into the cathode at a flow rate of 100 mL min^−1^. A constant temperature of 30 °C or 60 °C was maintained using heating tapes adhered to the end plates and controlled through a software controller with cell temperature feedback from a thermocouple inserted into a flow field plate. An electrical load (PLZ164WA electrochemical system, Kikusui Electronics Corporation, Tokyo, Japan) was used to record the current density (I) and potential (V) values. The maximum peak-power density (
$P_{\max}$
) was determined from the maximum power (V and I product) value.

## 3. Results and Discussion

### 3.1. Morphological and EDX Mapping Analysis

A TEM micrograph of pure ZIF-8 nanoparticles shows a cubic shape (50 to 80 nm), as displayed in Figure 1a. The Brunauer–Emmett–Teller (BET) surface area (Appendix A) and Langmuir surface area of ZIF-8 nanoparticles show 1430 m^2^ g^−1^ and 191 m^2^ g^−1^, respectively. The single point total pore volume was 1.02 cm^2^ g^−1^. The Barrett–Joyner–Halenda (BJH) adsorption average pore diameter and average pore width of the ZIF-8 particles were 21.8 and 28.6 nm. These results confirm ZIF-8 to be a porous framework structure. The surface area and pore diameter of the ZIF-8 particles matched the description reported in the literature [35,36]. 

The cross-sectional images of different ZIF-8 loadings in the PVA/ZIF-8/GA composites are displayed in Figure 1. The PVA/GA shows a dense and smooth surface morphology without any cracks, as shown in Figure 1b. Photographic images of PVA/GA also display transparent smooth film (Appendix A). The low and higher magnified FESEM (Figure 1c–f) shows a uniform ZIF-8 nanofiller distribution in the polymeric matrix by increasing the ZIF-8 loadings from 25.4% to 40.5%. Moreover, no voids or defects are shown in the composite membrane between the nanofillers and polymer matrix. This high ZIF-8 load was realized using the water-phase preparation method. Amirilargani and Sadatnia reported that 7.5–10% of dry ZIF-8 nanofillers blended with PVA matrix lead to metal organic framework aggregation and decreased composite compactness [37]. In this work, the ZIF-8 nanoparticle aqueous solution without drying was directly mixed with the polymer. The ZIF-8 particles were hydrated using water molecules to distribute homogeneously in the polymeric matrices with up to 40.5% content. This was due to the excellent interfacial compatibility between water-based ZIF-8 nanofillers and polymeric matrix in the composites. Increasing the ZIF-8 nanofillers concentration (45.4%), the ZIF-8 particle non-homogeneous distribution in the polymeric matrix was observed (Figure 1g,h). Appendix A shows the corresponding surface photographic images of PVA/40.5%ZIF-8/GA and PVA/45.4%ZIF-8/GA composite membranes. The PVA/45.4%ZIF-8/GA membrane had some white particles, which might be due to ZIF-8 aggregates.

The corresponding EDS mapping micrographs of Zn element distribution in the composites with various ZIF-8 nanofiller contents are shown in Appendix A. Zn represents the existence of ZIF-8 particles in the composites. They confirmed that the ZIF-8 particles were mixed well and were evenly distributed in the composite membranes. Additionally, the Zn weight percentages increased (8.56% to 15.5%) with increasing ZIF-8 nanofiller contents, as shown in Appendix A. This indicates that the composites were homogeneous at the micrometer scale.

### 3.2. Structural Analysis of PVA/ZIF-8/GA Composites

The X-ray diffraction (XRD) patterns of ZIF-8 nanoparticles, PVA/GA, and the PVA/ZIF-8/GA composites are represented in Figure 2a. The strong XRD peaks observed at 7.3° (011), 10.3° (002), 12.7° (112), 14.8° (022), 16.4° (013), 18.0° (222), 24.5° (233), and 26.8° (134) were attributed to the ZIF-8 particles [38]. Increasing the ZIF-8 content in the PVA matrix, the intensities of the corresponding ZIF-8 peaks increased in the composite membrane without peak shifts. Moreover, the ZIF-8 phase structure was well preserved after being mixed with PVA polymeric solution. 

The FTIR spectra of the pure PVA and GA-crosslinked PVA membranes are represented in Figure 2b. The pure PVA spectrum displayed sharp peaks at 1088 and 1332 cm^−1,^ represented by the stretching and deformation vibration of C–O–C groups. The peak at 1650 cm^−1^ is attributed to the stretching vibration of -OH groups that are sensitive to intermolecular interaction [39] or C-H bending vibration in PVA [40]. In addition, the peak at 1707 cm^−1^ denotes the stretching vibration of carbonyl (C=O) groups from the existence of acetate groups [41]. The characteristics vibration at 2860 and 2928 cm^−1^ attributed to the asymmetric and symmetric stretching vibration of C-H from alkyl groups [25]. The broad peak at 3328 cm^−1^ represents the axial deformation of the -OH groups involved in PVA intra- and intermolecular hydrogen bonding [42,43]. In the case of the PVA/GA membrane, the decreasing FTIR peak intensity is due to the formation of ether linkages and acetal rings [25]. This resulted from the reaction between the OH groups of PVA and the aldehyde groups of GA [40,44]. 

The high-resolution C1s XPS spectra of PVA (prepared according to [31]), PVA/GA, and PVA/40.5%ZIF-8, without and with GA-crosslinked membranes, are shown in Figure 2c–f. The XPS peaks of the PVA at 284.5, 285.8, and 289 eV were assigned to C-C (47%), C-O (49%), and C=O (3%) bonds, respectively, and correspond with the reported literature [45]. After GA-crosslinked with PVA, the C-C (45%) bond intensity decreased and increased the hydrophilic groups of C-O (51%) and C=O bonds (4%), as shown in Figure 2d. Moreover, the C1s peak shifted towards higher binding energies in the PVA/GA sample. The aldehyde groups participated in the reaction with hydroxyl groups in PVA polymeric chains. Thereby, the C-O/C=O bond of acetal and ether linkage percentage increased, which confirms the successful GA crosslinking with PVA segments. After the addition of ZIF-8 nanofillers in the PVA or PVA/GA samples, only the C-C and C-O bonds were visible, and the C=O bond disappeared (Figure 2e,f) due to complete PVA alcoholysis [45]. This might be attained with higher nanofiller loadings (40.5%) and ZIF-8 particle interfacial bonding/interaction with PVA segments. The C-O percentage of PVA/ZIF-8/GA (47%) was increased higher than that of the PVA/ZIF-8 (44%) composite due to increased bond stability between the aldehyde groups and PVA polymeric matrix. 

### 3.3. Mechanical Analysis of Nanocomposite Membranes

The mechanical behaviors of PVA/ZIF-8/GA composites (0%, 25.4%, 40.5%, and 45.4% ZIF-8 loadings) are presented in Figure 3 and Table 1. With the addition of ZIF-8 into the PVA matrix, the composite membranes significantly lost ductility, and the elongation was reduced by two orders of magnitude at 25.4–45.4% ZIF-8 loads (Table 1). The tensile strength (5.65 to 7.49 MPa) and Young’s modulus (29.5 to 313.3 MPa) were increased from pure crosslinked PVA to 40.5% ZIF-8 load but decreased at 45.4% loading, as shown in Table 1. At this high ZIF-8 load, the composite became brittle. Compared to other MMMs incorporated with dry fillers [37,38], this water-based synthesis and composite preparation technique can tolerate higher filler concentration. Thereby, the mechanical properties of these composites were higher than those reported in the literature on PVA@10%ZIF-8 composite (4.5 MPa vs. 21 MPa) [46].

### 3.4. Crystallinity, Alkali Uptake, and Ionic Conductivity 

The polymeric crystalline nature of PVA/ZIF-8/GA composites were determined using DSC, and the data are represented in Table 1. All samples exhibited endothermic peaks at 215–225 °C. The PVA/GA polymer showed a crystallinity of 36.5%, which was slightly lower than that of pure PVA film (38.1% [31]). The chemical crosslinking of GA with PVA chains reduced the hydroxyl groups in the PVA and weakened hydrogen bond formation, thereby decreasing the chain packing and the degree of crystallinity [44]. When increasing the ZIF-8 nanofiller loadings in the PVA/GA matrix, the polymeric crystallinity of the composite membranes decreased. The polymeric crystallinity of PVA/40.5%ZIF-8/GA was decreased to 29.15%, which was the lowest (Table 1) among the tested samples. The amorphous region was increased with the addition of ZIF-8 because the fillers inhibited the polymeric backbones from aligning [16]. Yang et al. also reported that higher montmorillonite ceramic nanofiller loads in the GA-crosslinked PVA showed an augmented amorphous phase [47], which is in line with our findings. The crystallinity of the PVA composite membrane depends on the PVA molecular weight. For the low-molecular-weight PVA (89–98 k Da) composites containing fumed silica fillers, the polymer crystallinity decreased from 51.5% to 44.1% and 41.4% at 20% and 30% filler loadings, respectively [18]. In the case of high-molecular-weight PVA (146–186 k Da) composites, the crystallinity decreased from 38.1% (pure PVA) to 34.2% at 25.4% nanofiller load [31]. The higher molecular-weight PVA exhibits longer chains and more chain entanglement and is less likely to form crystallites. The increased amorphous region and free volume properties are beneficial for ionic conduction [48]. 

The swelling resistance in alkaline solution is an important factor to determine the stability of AEMs. The PVA/GA membrane shows higher swelling ratio of 51.5%. Incorporating 25.4% ZIF-8 into the polymeric matrix, the volume swelling behavior slightly decreased (swelling ratio of 49%). Increasing the ZIF-8 loads to 40.5–45.4% exhibited negligible volume swelling changes (<1.4%) and confirmed the dimensional stability. Much literature used dried ZIF nanofiller powders (1–10 wt.%) into polymeric matrices and obtained moderate to high swelling ratios (22–320%) [49,50,51,52]. Such highly swollen composites may suffer from dimensional changes and weakened mechanical strength. In this study, the ZIF-8 nanoparticles were synthesized using the water-based method [30], and the as-prepared aqueous suspension was directly added to the hydrophilic PVA solution. This improved the chemical compatibility of the solid fillers and the polymer, resulting in the homogeneous nanofiller distribution (as shown in Figure 1) in the composite. Such uniformity was affixed with the help of the GA crosslinker to suppress polymer chain mobility. The combined water-base ZIF-8 synthesis and GA crosslinking approaches result in good swelling resistance, which is beneficial for alkaline fuel cell applications. 

The PVA/GA membrane absorbed 0.81 g g^−1^ KOH molecules per weight of the polymer (Table 1). The alkali uptakes of the GA-crosslinked PVA samples were slightly lower than that for the un-crosslinked sample (0.81 vs. 0.92 g g^−1^, Table 1 and [31]). The crosslinked chains lost flexibility, and swelling was prohibited. Increasing the ZIF-8 nanofiller content in the PVA polymeric matrix, the adsorbed KOH in the composites increased from 0.89 to 1.08 g g^−1^ (Table 1). This was because of the enhanced polymeric amorphous region in the PVA/ZIF-8/GA composite membranes, which enabled intermolecular interaction and promoted KOH uptake. It has also been reported that nanofiller addition to the composite membrane played an important role in preventing polymer chain alignment and ion diffusion [53]. 

The membrane resistance at 30 °C could be measured with an AC impedance analyzer, and the through-plane conductivity is summarized in Table 1. The ionic conductivity of PVA/GA was 1.15 × 10^−2^ S cm^−1^ and 1.35–1.41 × 10^−2^ S cm^−1^ for the PVA/ZIF-8/GA composites. The improvement in ionic conductivity by increasing ZIF-8 nanofiller content resulted from the higher alkali uptakes and ionic transport rates within the polymer electrolytes. 

### 3.5. Alcohol Permeability of Nanocomposite Membranes

The methanol permeability of PVA/ZIF-8/GA composites (0%, 25.4%, 40.5%, and 45.4% ZIF-8 loadings) are studied using 2 M methanol. The results are represented in Table 1. The methanol permeability of PVA/GA was 9.6 × 310^−6^ cm^2^ s^−1^, whereas the permeability decreased with increasing ZIF-8 nanofillers concentrations in the GA-crosslinked PVA up to 40.5% load (0.54 × 10^−6^ cm^2^ s^−1^). This confirms that the diffusion coefficient or the solubility of methanol was reduced by increasing the ZIF-8 content in the PVA/GA matrix [31]. The increased polymeric amorphous region at a higher ZIF-8 load facilitated water and hydroxide ion passage but limited methanol transport. We reported that PVA composites demonstrate increased free volume upon filler addition as the result of both increased free volume hole intensity and size. However, the enlarged free volume hole size was not sufficient for alcohol passage [53]. In addition, the ZIF-8 pore size was 0.34 nm [54], which might permit hydroxide (ionic radius of 0.11 nm [55]) and water molecules (kinetic diameter of 0.296 nm [56]) to pass through while blocking methanol (kinetic diameter of 0.38 nm [56]).

Note that as the ZIF-8 content was increased to 45.5%, the methanol permeability was raised to 1.12 × 10^−6^ cm^2^ s^−1^. The filler aggregates (shown in previous sections) form voids or cracks, allowing methanol molecules to breakthrough. Nevertheless, this load was much higher than those of other fillers incorporated into the polymeric matrix. The allowed filler load is discussed in the last section.

The methanol permeability data for the un-crosslinked PVA/ZIF-8 composites [31] were adopted for comparison. Because the crosslinked PVA showed a lower polymer crystallinity (due to chain confinement) than the un-crosslinked pure PVA film (36.5% vs. 38.1% [31]), the methanol was transported faster in the crosslinked sample (9.63 × 10^−6^ cm^2^ s^−1^ vs. 4.28 × 10^−6^ cm^2^ s^−1^ [31]). Upon filler addition, the permeability trend started to shift. In the 25.4% ZIF-8 composites, the crosslinked and un-crosslinked composites showed comparable methanol permeabilities (1.66 × 10^−6^ cm^2^ s^−1^ vs. 1.48 × 10^−6^ cm^2^ s^−1^ [31]), owing to confined chain movement in the chemical crosslinker (GA) and physical steric hindrance (nanofillers). As the filler content was raised to 40.5% and 45.4%, the methanol permeability was further suppressed in the crosslinked composites, and the methanol transport rates were only half those of their un-crosslinked counterparts. 

### 3.6. Selectivity of Nanocomposite Membranes

The conductivity-to-permeability selectivity of the GA-crosslinked PVA/ZIF-8 composites are represented in Figure 4 and Table 1. The values of un-crosslinked samples in Figure 4 are taken from previous work [31]. At low nanofiller loadings (0–25.4%), both un-crosslinked and GA-crosslinked samples have a similar range of selectivity values. When increasing the ZIF-8 nanofiller concentrations to 40.5% and 45.4%, the selectivity of GA-crosslinked composites increased to 1.5–2 fold than those of the un-crosslinked samples. The selectivity reached the highest value at 40.5% ZIF-8 loading, primarily due to the lowest permeability. 

Yang et al. [47] reported that 15–20% of montmorillonite-ceramic-nanofillers-loaded GA-crosslinked PVA samples exhibited lower ionic conductivity than the composite with <10% loading. This was due to poor dispersion (chunks) or large aggregation of nanofillers in the polymeric matrix restricting the ionic transport pathways. It was also reported that incorporating >5% of nanofillers in the PVA polymer matrix leads to high filler agglomeration due to poor nanofiller dispersion capability and obtained non-uniform composite for alkaline fuel cells [57,58]. Remarkably, the present water phase technique shows improved selectivity due to the uniform distribution of ZIF-8 nanofillers in the polymeric matrix, even at higher concentrations (up to 40.5% of ZIF-8 loadings). The results further confirmed that the aqueous phase method could overcome the nanofiller aggregation problem in the MMMs for high-filler loadings, as compared with solid nanofiller incorporation methods [47,59,60].

### 3.7. Alkaline Stability of Nanocomposite Membranes

The PVA/GA and the PVA/40.5%ZIF-8/GA composites were placed in 6 M alkaline solution for various time periods to measure the long-term alkaline stability in terms of the through-plane ionic conductivity. The samples were evacuated at 30 °C after immersion in 6 M alkaline solution for 24 h and 168 h to observe the change in ionic conductivity, as shown in Figure 5. After 168 h in alkaline solution, the PVA/GA membrane had 3.04% ionic conductivity loss when compared with the initial conductivity. With the 40.5% ZIF-8 content in the PVA matrix, the ionic conductivity showed almost negligible change (0.19%) during the 168 h in the alkaline solution. The addition of ZIF-8 nanoparticles resulted in a compact physical network structure among the PVA/GA molecular chains, decreasing the swelling of PVA chains and molecular relaxation. Moreover, the present work of GA-crosslinked samples maintained ionic conductivity, as compared with those counterparts without crosslinking [31]. Without GA crosslinking, the pristine PVA and PVA/40.5%ZIF-8 composites showed drastically decreased conductivity (31.28% and 14.14% loss [31]). The loss of ionic conductivity was owing to less KOH being retained on the membrane and the weakening of the backbone of the polymeric matrix [22,34,61]. With the addition of a GA crosslinker, the polymer chain dissolution was inhibited, and the composite membrane confined structure was able to retain KOH in the PVA matrix and maintain strong stability in alkaline.

### 3.8. Single-Cell Performance of PVA/ZIF-8/GA Composites

The DAMFC performance of single cells using the PVA/GA and PVA/ZIF-8/GA composites at 30 °C and 60 °C is displayed in Figure 6a–h. The composites containing ZIF-8 loads of 0%, 25.4%, 40.5%, and 45.4 % resulted in open-circuit voltage (V_oc_) values of 0.69, 0.61, 0.68, and 0.49 V, respectively, at 30 °C. Their resultant maximum power densities (P_max_) were 40.5, 80.3, 81.7, and 66.3 mW cm^−2^, respectively. The DAMFC performance using the PVA/40.5%ZIF-8/GA composite exhibited the highest V_oc_ and P_max_ among the tested samples. As the temperature was raised to 60 °C, the V_oc_ (0.80, 0.69, 0.76, and 0.58 V) and P_max_ (67.7, 178.4, 190.5, and 150.2 mW cm^−2^) values were improved. The higher operating temperature could promote electrochemical kinetic reactions [62]. Moreover, the ionic conductivity was higher with increasing temperature (Table 1). The slope of the ohmic loss region was lower at the higher temperature, and the cell was able to sustain itself at higher current densities (Figure 6a–d vs. Figure 6e,f).

The DAMFC performance using composites of different ZIF-8 contents in the PVA/GA at 30 and 60 °C is displayed in Figure 7. At both temperatures, the cell voltage and power density were increased with increasing ZIF-8 loads until 40.5%. The PVA/40.5%ZIF-8/GA composite shows the highest P_max_ of 82 mW cm^−2^ and 191 mW cm^−2^ at 30 °C and 60 °C. Increasing the temperature could improve the P_max_ values of all the samples due to the faster electrochemical kinetic reaction rates and lower electrical resistance in the single fuel cell [28,63]. The optimal load of ZIF-8 nanoparticles was 40.5%, which was associated with the highest selectivity, as shown in Figure 5. When the ZIF-8 load was increased further to 45.4%, the power density decreased. This was due to the defects caused by the non-homogeneous ZIF-8 distribution in the composite (Figure 1h). This resulted in a high methanol cross-over rate. Similar behavior was also reported when increasing nanofillers load in the PVA/CNT composite [63]. The reduced conductivity-to-methanol permeability ratio (selectivity in Figure 4) was parallel to the reduced P_max_ values. The miscibility of aqueous ZIF-8 and PVA solutions seems to reach an optimal level at 40.5% ZIF-8 load. This high ZIF-8 load was not easily achieved conventionally by mixing PVA solutions with dry filler particles because fillers tend to form aggregates in the MMMs [53].

The PVA/40.5%ZIF-8/GA resulted in a higher P_max_ by 10% than the uncrosslinked PVA/40.5%ZIF-8 composite (190.5 vs. 173.2 mW cm^−2^ [31]) at 60 °C. The pre-crosslinking treatment was beneficial for good membrane characteristics and cell performance. In our previous work [26], QPVA/GO-Fe_3_O_4_ was cast and dried, and the film was post-crosslinked with GA. Although the post-crosslinking treatment enhanced the polymer amorphous phase and mechanical stability, the fuel cell performance was sacrificed [26]. It seems that the pre-crosslinking step before the aqueous ZIF-8/PVA mixture water phase is preferred as this process saves energy to dry ZIF-8 and GA can fully crosslink the polymer chains prior to film casting.

### 3.9. Fuel Cell Performance Comparison

Table 2 summarizes our DAMFC peak-power density literature data from the fuel cells using MMMs and their corresponding pristine membranes [17,26,29,31,62,63,64,65,66,67,68,69]. It is very clear that incorporating nanofillers into a polymeric matrix to form fuel cell electrolytes can significantly improve power generation. We reported that these fillers provide steric hindrance to interrupt semi-crystalline polymer chain alignment, generating increased polymeric fractional free volumes (FFV) at the expense of polymer crystallinity. For the PVA/fumed silica (FS) composites with 0–30% loads, the FFV increased from 1.7 to 2.8% [48] and crystallinity decreased from 51 to 41% [18]. Even at a minute amount (0.05%) carbon nanotube (CNT) load, the PVA/CNT membrane exhibited an FFV of 3.53%, increased from 2.48% of pure PVA film [63]. At the same time, the polymer crystallinity dropped from 42% to 38% [70]. For crosslinked quaternized PVA/quaternized chitosan (QPVA/Qchitosan) composite at 5% load, the FFV increased from 2.0 to 3.1%, and crystallinity decreased from 9.8 to 6.4% [17,66]. In the present work, the FFV of the crosslinked composite increase from 4.45% (without ZIF-8 addition) to 8.36% is accompanied with a crystallinity decrease from 37 to 29% (Table 1).

Detailed examination of the FFV characteristics using positron annihilation lifetime spectroscopy reveals that these composites demonstrated higher free volume hole densities than their pure polymer counterparts [17,48], contributing to the increased FFV. The free volume hole size may be slightly enlarged in the PVA/FS or shrunk in the crosslinked QPVA/Qchitosan composites. These hole sizes (radius of 2.2–2.7 Å [17,48]) can provide sufficient pathways for hydroxide ions (radius of 1.1 Å) and water diffusion [69] but not for larger methanol penetrant. Consequently, the nanocomposite ionic conductivities were improved, whereas the fuel cross-over rates were prohibited. The overall effect of adding nanofillers into polymers was the reduced ohmic losses shown in the polarization curves and increased peak-power densities.

Note that optimal nanofiller loads strongly depend on the filler geometry and the compatibility with the base polymer. For zero-dimension fillers, such as FS and Qchitosan, 20–30% loads are probably reaching the upper boundary, and higher loads beyond this threshold tend to form cracks and defects due to dispersion heterogeneity. For one- and two-dimensional nanofillers (e.g., CNTs, GO, etc.), large surface areas allow much interaction (van der Waals force, electrostatic interactions, hydrogen bonding, ion-dipole, etc.) with polymers [40,71], and the optimal loads are only a few percent or even tenths of a percent. This water-based synthesis limited the aggregation problems, even at higher nanofiller loads (40.5%), and improved the homogeneous distribution of ZIF-8 nanofillers in the polymeric matrix without any cracks or defects (Figure 1). The optimized PVA/40.5%ZIF-8/GA composite shows 181% P_max_ improvement, as compared with the pristine PVA/GA sample. The attained P_max_ values in this work were higher than those previously reported for GA-crosslinked electrolytes (Table 2) under similar operating conditions. 

## 4. Conclusions

In this work, the water-based synthesis method was used to prepare ZIF-8 nanoparticle suspension, which was mixed directly into a PVA solution (without an intermediate particle drying step) to achieve uniform nanofiller dispersion. This structure was retained using the GA crosslinker. This approach produces composites containing high ZIF-8 concentrations (up to 40.5%). The ZIF-8 nanofiller homogeneity in the composites and defect- and crack-free structure were clearly observed using FESEM and EDS analyses. The combination of uniform nanofiller distribution and crosslinker affixation improved swelling resistance, maintained polymer chain confinement, retarded methanol cross-over, retained mechanical robustness, and prevented conductivity decay of the composites.

An increased polymeric amorphous region in the composite was reported and positively correlated to KOH uptake and ionic conductivity. Methanol permeability was suppressed in the composites and reached a minimum at the highest (40.5%) ZIF-8 load. The resultant selectivities of GA-crosslinked composites were 1.5–2 times higher than those of the un-crosslinked samples. The higher ZIF-8 loads (40.5%) in the crosslinked composite exhibited negligible volume swelling ratios and good dimensional stability. The optimal ZIF-8 load was 40.5% in the composite, and the achieved P_max_ was 190.5 mW cm^−2^ at 60 °C. The obtained P_max_ value is 181% higher than that of pure PVA/GA membrane. These swollen-resistant and stable solid electrolytes are promising in alkaline fuel cell applications.

## Figures and Tables

**Figure 1 nanomaterials-12-00865-f001:**
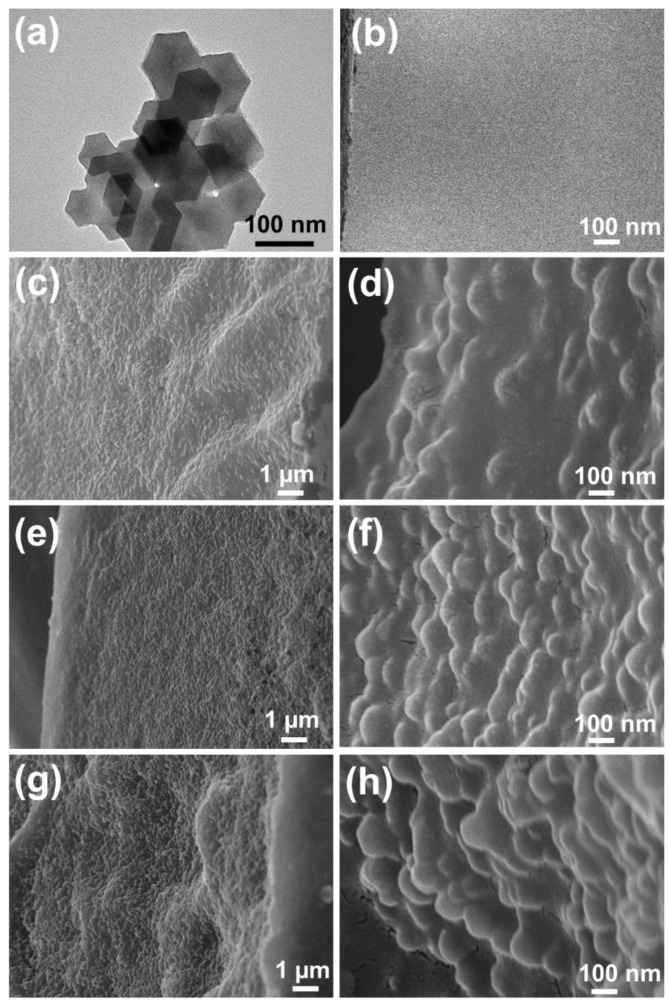
(**a**) Transmission electron microscopic image of pristine ZIF-8 nanoparticles; low and high magnification cross-sectional FESEM images of (**b**) PVA/GA, (**c**,**d**) PVA/25.4%ZIF-8/GA, (**e**,**f**) PVA/40.5%ZIF-8/GA, and (**g**,**h**) PVA/45.4%ZIF-8/GA composite membranes.

**Figure 2 nanomaterials-12-00865-f002:**
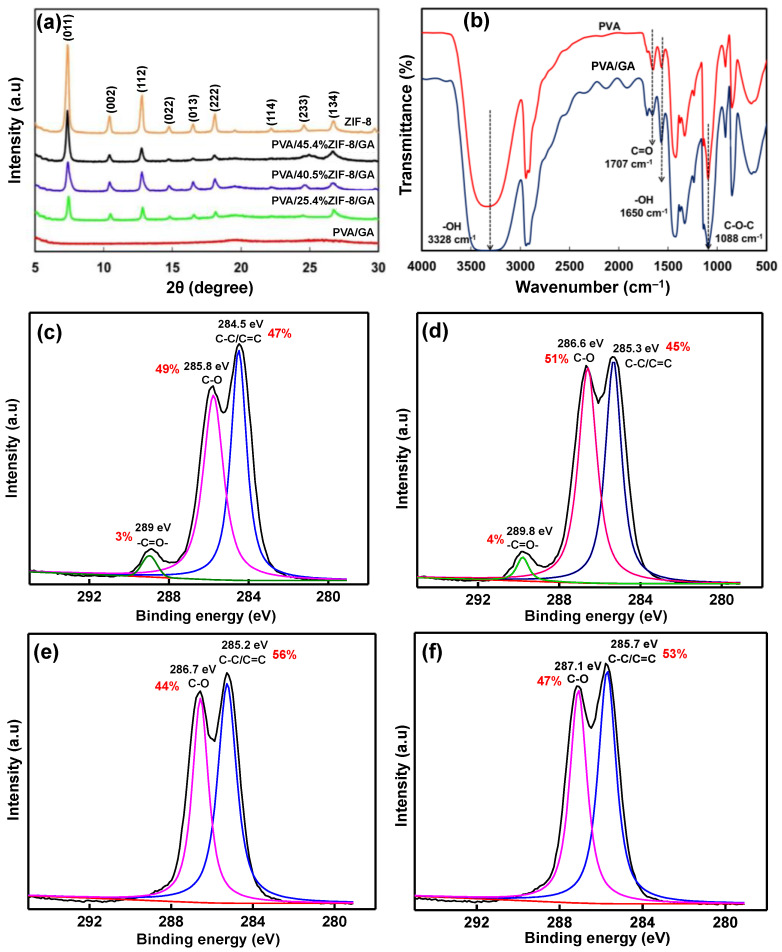
(**a**) X-ray diffraction patterns of ZIF-8 nanoparticles and PVA/GA and PVA/ZIF-8/GA composites (25.4%, 40.5%, and 45.4% ZIF-8 loadings); (**b**) FTIR spectra of pure PVA membrane (prepared as in [31]) and GA-crosslinked PVA composite; and XPS detailed scans of C1s of (**c**) PVA, (**d**) PVA/GA, (**e**) PVA/40.5%ZIF-8 (prepared as in [31]), and (**f**) PVA/40.5%ZIF-8/GA composites.

**Figure 3 nanomaterials-12-00865-f003:**
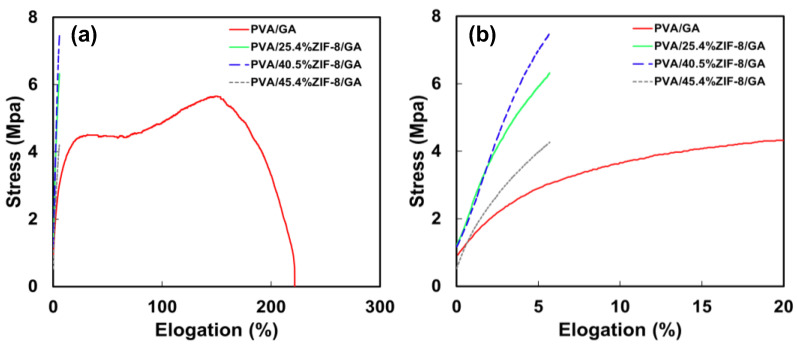
Stress-strain curves of PVA and PVA/ZIF-8 with GA crosslinking in (**a**) entire and (**b**) low elongation ranges.

**Figure 4 nanomaterials-12-00865-f004:**
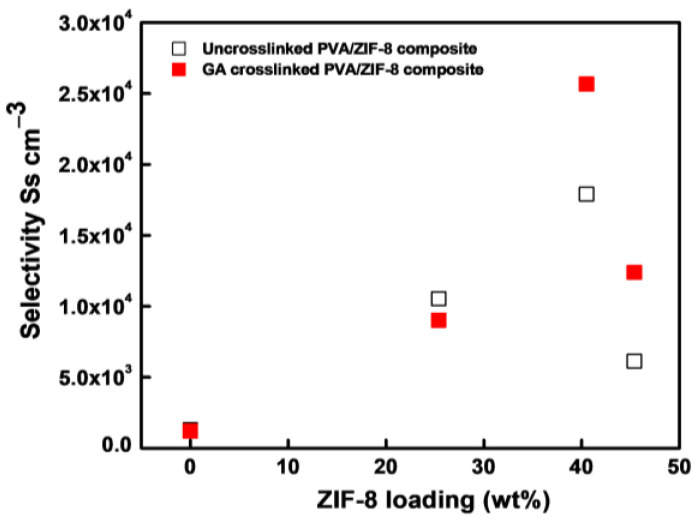
Selectivity of un-crosslinked (data from [31]) and GA-crosslinked PVA/ZIF-8 composites (0%, 25.4%, 40.5%, and 45.4% ZIF-8 loadings) at 30 °C.

**Figure 5 nanomaterials-12-00865-f005:**
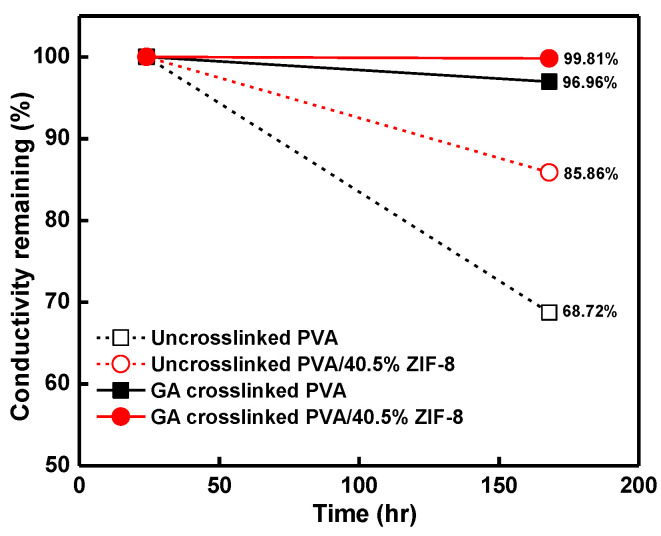
Relative through-plane conductivity decay of un-crosslinked (data from [31]) and GA-crosslinked PVA and PVA/40.5%ZIF-8 composites after 168-h alkaline treatment time.

**Figure 6 nanomaterials-12-00865-f006:**
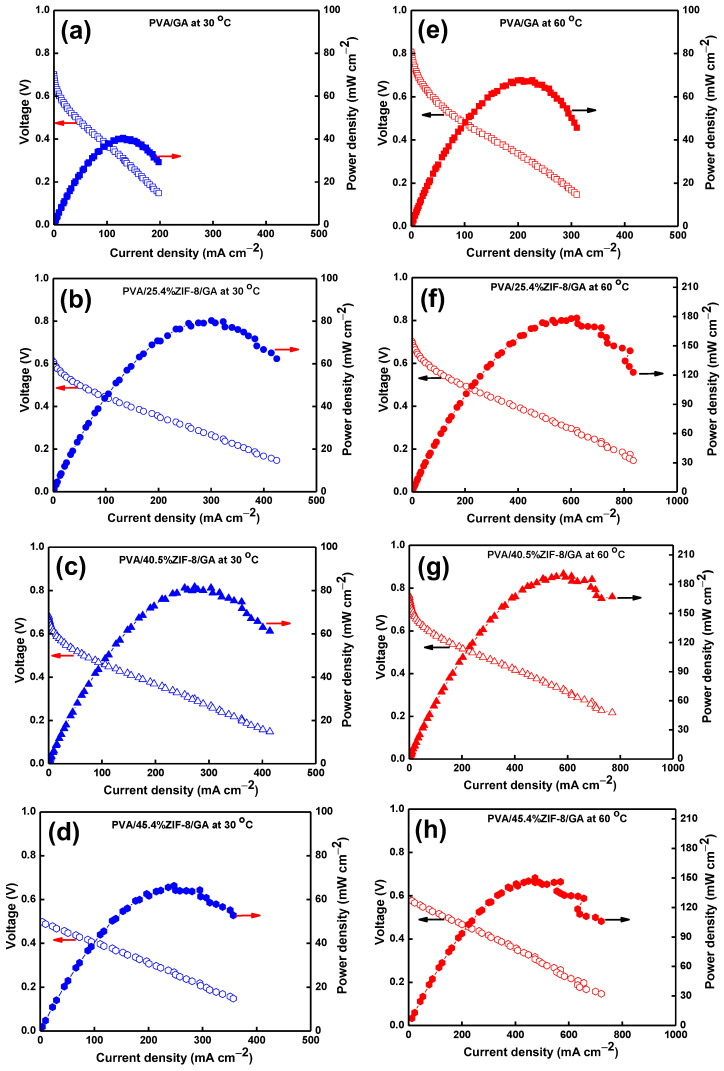
Single-cell performance using PVA/GA, PVA/25.4%ZIF-8/GA, PVA/40.5%ZIF-8/GA, and PVA/45.4%ZIF-8/GA composite electrolytes: (**a**–**d**) cell voltage and power density at 30 °C, and (**e**–**h**) cell voltage and power density at 60 °C. The open symbols marked with left arrows show cell voltage data (value and unit in left Y axis) and the closed symbols with right arrows correspond to power density values (right Y axis).

**Figure 7 nanomaterials-12-00865-f007:**
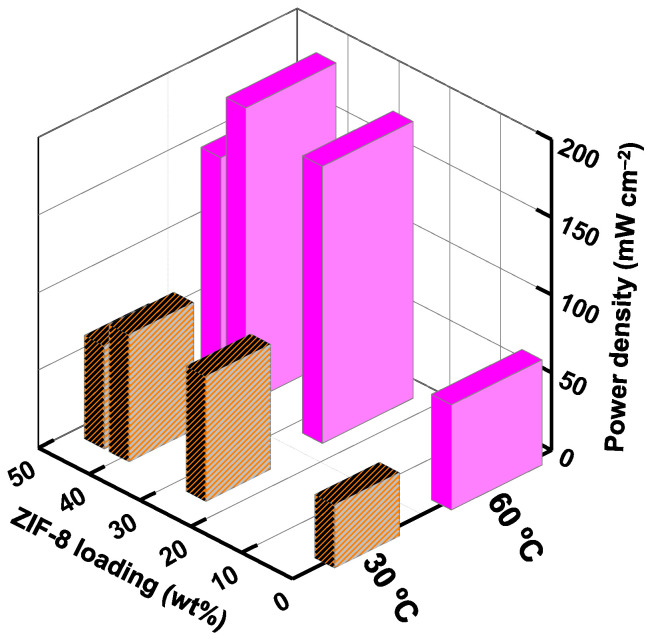
Peak-power density comparison of PVA/ZIF-8/GA composites (0%, 25.4%, 40.5%, and 45.4% ZIF-8 loadings) at 30 °C and 60 °C.

**Table 1 nanomaterials-12-00865-t001:** Structural, mechanical, and electrochemical properties of the PVA/GA and the PVA/ZIF-8/GA composites at 30 °C.

ZIF-8 Load in Membrane	0%	25.40%	40.50%	45.40%
Tensile strength (MPa)	5.65	6.32	7.49	4.27
Elongation (%)	228	5.69	5.69	5.69
Young’s modulus (MPa)	29.5	264	313	123
Polymer crystallinity (%)	36.5	30.2	29.1	29.4
Alkali uptake (g g^−1^)	0.81	0.89	0.98	1.08
Through-plane ionic conductivity (×10^−2^ S cm^−1^)	1.15	1.35	1.39	1.41
Methanol permeability (10^−6^ cm^2^ s^−1^)	9.63	1.66	0.54	1.12
Selectivity (Ss cm^−3^)	1190	9020	25,700	12,400

**Table 2 nanomaterials-12-00865-t002:** Peak-power density (P_max_) comparison of DAMFCs using polymeric composites and pure membranes (data in parentheses under P_max_ column), reported from the authors’ group at 60 °C.

Electrolytes	Filler Loading (wt.%)	Peak-power Density (P_max_) (mW cm^−2^)	Pmax Increment Compared with Pure Sample (%)	Source
PBI/GO spin coated	0.6	140 (110)	27	Yu et al. [62]
PBI/GO	1	231 (225)	2.7	Chang et al. [67]
PBI/GO-Fe_3_O_4_	0.2	176 (145)	21	Kumar et al. [68]
PVA/CNTs	0.05	39 (27)	45	Pan et al. [69]
PVA/CNTs	0.05	39 (20)	95	Lue et al. [63]
QPVA/Qchitosan	5	73 (38)	92	Liao et al. [17]
QPVA/fumed silica	5	88 (36)	146	Kumar et al. [65]
QPVA/CTAB coated LaFeO_3_	0.1	272 (155)	76	Kumar et al. [29]
QPVA/GO-Fe_3_O_4_/GA with magnetic field ^a^	0.1	55 (22)	147	Lin et al. [26]
QPVA/Chitosan/GA ^b^	10	58 (40)	46	Li et al. [64]
QPVA/Qchitosan/GA ^b^	5	73 (40)	83	Liao et al. [66]
QPVA/Qchitosan/GA ^b^	20	50 (40)	25	Liao et al. [66]
PVA/ZIF-8	40.5	173 (81)	114	Hsu et al. [31]
PVA/ZIF-8/GA ^c^	40.5	191 (68)	181	This work

^a^ Membrane: post-crosslinked; Pt/C (5 mg cm^−2^); Pt-Ru/C (6 mg cm^−2^). ^b^ Membrane: pre-crosslinked; Pt/C (5 mg cm^−2^); Pt-Ru/C (6 mg cm^−2^). ^c^ This work: pre-crosslinked; Pt/C (1 mg cm^−2^); Pt-Ru/C (2 mg cm^−2^). GA: glutaraldehyde. QPVA: quaternized PVA. CNTs: carbon nanotubes. Qchitosan: quaternized chitosan particles. PBI: polybenzimidazole. GO: graphene oxide. GO-Fe_3_O_4_: graphene oxide decorated with Fe_3_O_4_ nanoparticles.

## Data Availability

Not applicable.

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
