# Peer review of "Swelling-Resistant, Crosslinked Polyvinyl Alcohol Membranes with High ZIF-8 Nanofiller Loadings as Effective Solid Electrolytes for Alkaline Fuel Cells"

_nanomaterials, 2022, doi:10.3390/nano12050865_

Round 1
Reviewer 1 Report
The paper titled, Swelling-resistant, crosslinked polyvinyl alcohol membranes with high ZIF-8 nanofiller loadings as effective solid electrolytes for alkaline fuel cells presented for publication in Nanomaterials journal focalized in the preparation of aqueous zeolitic imidazolate frame work-8 into a polyvinyl alcohol as composite membranes. This work is interesting but the manuscript in its present form is not suitable for publication and I suggest major mandatory revision before further reconsideration.
- In the abstract, authors could present carefully the novelty of this work.
- The title contains the term swelling, however no discussion about this property
- Experimental part: Volumes, concentrations, weights of all chemicals should be added
- Please revise the equation (1)
- Table S1 contains the most interesting results and it should be moved to the manuscript
- FTIR analysis (line 202-207) should rewritten and the results could be discussed with references
- Please be sure that is transmittance and not absorbance (line 205).
- I don’t see any discussion on the swelling properties of the membrane materials? Please add a section to evaluate the swelling properties which can be important for this work.
- Please there are 8 references form Yang, C.-C et al., please reduce the number to 3 max (if possible replace them by other from this journal)
Author Response
Reviewer 1
The paper titled, Swelling-resistant, crosslinked polyvinyl alcohol membranes with high ZIF-8 nanofiller loadings as effective solid electrolytes for alkaline fuel cells presented for publication in Nanomaterials journal focalized in the preparation of aqueous zeolitic imidazolate frame work-8 into a polyvinyl alcohol as composite membranes. This work is interesting but the manuscript in its present form is not suitable for publication and I suggest major mandatory revision before further reconsideration.
Response: Thank you for your suggestions and positive comments.
Q1. In the abstract, authors could present carefully the novelty of this work.
Response: The novelty of this work is included in the abstract, and the last paragraph of the Introduction.
Q2. The title contains the term swelling, however no discussion about this property
Response: The new experimental data on the volume swelling is added and its discussion is included in the 2nd paragraph of the section 3.4, p. 8.
Q3. Experimental part: Volumes, concentrations, weights of all chemicals should be added.
Response: Thank you for pointing this out. All details are included in the section 2.2, p. 3.
Q4. Please revise the equation (1).
Response: Thank you for pointing this out. The typo error is corrected in the section 2.3, p. 4.
Q5. Table S1 contains the most interesting results and it should be moved to the manuscript.
Response: Thank you for your suggestion. The content in Table S1 was removed and the data values are included in the 1st paragraph of section 3.1, p. 4.
Q6. FTIR analysis (line 202-207) should rewritten and the results could be discussed with references.
Response: The FTIR section is revised with suitable supporting references (2nd paragraph of section 3.2).
Q7. Please be sure that is transmittance and not absorbance (line 205).
Response: The entire FTIR paragraph section is revised.
Q8. I don’t see any discussion on the swelling properties of the membrane materials? Please add a section to evaluate the swelling properties which can be important for this work.
Response: Thank you for your suggestion. The new experiment data on the volume swelling is added and its discussion is included in 2nd paragraph of section 3.4, p. 8.
Q9. Please there are 8 references form Yang, C.-C et al., please reduce the number to 3 max (if possible replace them by other from this journal).
Response: Thank you for your suggestion. We reduced the C.C. Yang et al., references to 3 and also included some MDPI papers (Section 3.6, p. 10).

Reviewer 2 Report
- Index the hkl values in XRD, in Fig. 2a
- Y-axis unit should be mentioned. There are many grammatical and typo errors throughout the manuscript.
- Cana authors prove, as single Figure of cell voltage and current density, in Fig. 6 for better understanding.
- Interpretation of results should be strengthened in order to meet the standard of journal and clear understanding of the subject.
- Fabrication of cell procedure should be given in more detail.
- Novelty of the work should be highlighted int eh introduction part. Some relevant reference should be included for strengthening the work, for example, Renewable & Sustainable Energy Reviews, 143 (2021) 110849; Ceramics International 47 (2021) 4404-4425.
Author Response
Reviewer 2
Q1. Index the hkl values in XRD, in Fig. 2a.
Response: Thank you for your suggestion. The hkl is indexed in Fig. 2a.
Q2. Y-axis unit should be mentioned. There are many grammatical and typo errors throughout the manuscript.
Response: Thank you for pointing this out. The Y-axis unit in Fig. 2b is mentioned. The typo errors were carefully checked and the grammatical errors were revised by a professional editor.
Q3. Can a authors prove, as single Figure of cell voltage and current density, in Fig. 6 for better understanding.
Response: Each graph in Fig. 6 is separated for clarity and revised in the manuscript.
Q4. Interpretation of results should be strengthened in order to meet the standard of journal and clear understanding of the subject.
Response: We revised the contents in the results and discussion section to improve clarity.
Q5. Fabrication of cell procedure should be given in more detail.
Response: Thank you for your suggestion. The details are included in the experimental section 2.4, p. 4.
Q6. Novelty of the work should be highlighted in the introduction part. Some relevant reference should be included for strengthening the work, for example, Renewable & Sustainable Energy Reviews, 143 (2021) 110849; Ceramics International 47 (2021) 4404-4425.
Response: The novelty of this work is included in the abstract, introduction and conclusion sections. The two references are included in the 1st paragraph of the introduction section.

Round 2
Reviewer 1 Report
Authors answered all my questions, and the revised paper is ready for publication as is.
Author Response
Q1. Authors answered all my questions, and the revised paper is ready for publication as is.
Response: Thank you very much for your positive feedback.
Q2. Response: The response to Editor is shown in the attached PDF file.
